# Cell Fusion in Human Cancer: The Dark Matter Hypothesis

**DOI:** 10.3390/cells8020132

**Published:** 2019-02-07

**Authors:** Julian Weiler, Thomas Dittmar

**Affiliations:** Chair of Immunology, Center for Biomedical Education and Research (ZBAF), Witten/Herdecke University, Stockumer Str. 10, 58448 Witten, Germany; julian.weiler@uni-wh.de

**Keywords:** cell fusion, cancer, metastasis, dark matter

## Abstract

Current strategies to determine tumor × normal (TN)-hybrid cells among human cancer cells include the detection of hematopoietic markers and other mesodermal markers on tumor cells or the presence of donor DNA in cancer samples from patients who had previously received an allogenic bone marrow transplant. By doing so, several studies have demonstrated that TN-hybrid cells could be found in human cancers. However, a prerequisite of this cell fusion search strategy is that such markers are stably expressed by TN-hybrid cells over time. However, cell fusion is a potent inducer of genomic instability, and TN-hybrid cells may lose these cell fusion markers, thereby becoming indistinguishable from nonfused tumor cells. In addition, hybrid cells can evolve from homotypic fusion events between tumor cells or from heterotypic fusion events between tumor cells and normal cells possessing similar markers, which would also be indistinguishable from nonfused tumor cells. Such indistinguishable or invisible hybrid cells will be referred to as dark matter hybrids, which cannot as yet be detected and quantified, but which contribute to tumor growth and progression.

## 1. Introduction

It is well known that cell–cell fusion and hybridization play a crucial role in several physiological processes, such as fertilization, placentation, myogenesis, osteogenesis, wound healing, and tissue regeneration. This process also occurs in cancers. However, its impact on cancer initiation and progression is as yet unclear (for review see [1,2,3,4,5]). This applies particularly to the question of whether cell fusion events do truly occur in human cancers and if the evolving tumor cell × normal cell hybrids and their progenies do truly contribute to disease progression, as was proposed by the German physician Otto Aichel in 1911 [6]. 

In fact, there have been a plethora of in vitro and in vivo studies in the past decades demonstrating that tumor cells do fuse with normal cells, such as macrophages, fibroblasts, stromal cells or stem cells, thereby giving rise to viable proliferating TN-hybrid cells with properties that are linked with tumor progression including enhanced tumorigenic and metastatic capacity or enhanced drug resistance [7,8,9,10,11,12,13,14,15,16,17,18,19,20,21,22,23,24,25,26,27,28,29,30,31]. Likewise, several studies have reported putative TN-hybrid cells in human cancers, in some cases comprising up to 40% of tumors [7,13,20,25,32,33,34,35,36,37,38,39,40,41,42]. Recently, Gast et al. showed that tumor × normal (TN)-hybrid cells could be found not only in human pancreatic ductal adenocarcinoma cells but also in the circulatory system where they were associated with a poor prognosis [29].

However, human TN-hybrid cells have been only identified in a few cancer types so far including breast [13,35], colorectal cancer [36], pancreatic cancer [29,42], melanoma [25,33,39], ovarian cancer [20], multiple myeloma [38], and renal cell carcinoma [32,34]. Hence, it remains unknown whether cell fusion is a common phenomenon that occurs in all cancers or if it is restricted to certain cancer types. Likewise, it remains unknown whether TN-hybrid cells that originate in the primary tumor contribute to tumor progression and metastasis formation. Some studies indicate that putative TN-hybrid cells can be found in metastases or in the circulation of cancer patients [7,29,33,34,39,41], but further studies are necessary to clarify whether circulating TN-hybrid cells are capable of inducing metastases. Finally, in some studies, TN-hybrid cells were identified by expression of hematopoietic markers, such as CD14, CD45, and CD163 [7,13,20,29,35,36]. While this is a relatively simple strategy for identifying putative TN-hybrid cells in human cancer biopsies, it cannot be ruled out that expression of macrophage-like antigens may be due to genomic instability, which is a hallmark of most, if not all, tumors and the main cause for intratumoral heterogeneity [43]. Genomic instability generates new mutations and/or gross chromosomal aberrations in dividing tumor cells [44]. This can be beneficial for the overall capacity of a tumor to adapt changes in its environment [44]. However, newly acquired genetic alterations can also compromise the genetic dominance of the tumor cells and, thus, affect tumor cell viability [44]. In this context, it should be noted that cell fusion is also a potent inducer of genomic instability. Hence, cell fusion can give rise to hybrids that may adapt better to changes in the tumor environment or to cancer therapy but can also give rise to nonviable hybrids. Likewise, hybrid cells may lose specific cell fusion markers over time as a result of genomic instability, thereby becoming indistinguishable from nonfused tumor cells.

Thus, to conclude that cell fusion and hybridization occurs between tumor cells and normal cells, highly specific markers are required to identify such hybrid cells, which is a tall order indeed. This brings us to the first question as follows.

## 2. What Would be Ideal Markers to Distinguish between TN-Hybrid Cells and Nonhybridized Tumor Cells?

A prerequisite in cell fusion research is to demonstrate that the cells truly fuse and hybridize with one another and that evolving TN-hybrid cells can be clearly identified. In some studies, cell lines were differentially labeled with various fluorochromes or fluorescent proteins such that they could be used to isolate putative hybrids [29,45,46,47,48,49,50]. In other studies, cells with drug resistance markers were used to isolate putative hybrids [16,17,47,51,52,53].

However, in in vivo tumor cell fusion studies, more complex strategies are necessary to show that, for example, cells of the hematopoietic lineage can fuse with tumor cells. It is interesting that some techniques, such as sex mismatch transplantation assays and parabiosis assays [54,55,56,57,58,59,60,61], were the same as those used successfully in hematopoietic stem cell (HSC)-based regeneration studies demonstrating that cell fusion gives rise to functional hybrid cells. In both assays, when cells from mouse A are transferred into mouse B, hybrid cells can be characterized by their morphology, functionality, sex chromosomes, and coexpression of mouse A and mouse B markers [54,55,56,57,58,59,60,61,62].

Using a sex mismatch transplantation assay, Rizvi and colleagues were able to show that transplanted bone marrow-derived stem cells (BMDCs) stably fused with normal and transformed intestinal epithelial cells [63]. Here, BMDCs from female EGFP or β-Gal transgenic mice were transplanted into lethally irradiated male APC^Min−/−^ mice, representing an animal model for colon carcinogenesis [63]. Similar findings were obtained using a parabiosis model, whereby a GFP mouse and a APC^Min−/−^ mouse, which was also transgenic for β-Gal, were surgically joined [12]. The intestinal crypts of the APC^Min−/−^ mouse were examined after seven weeks, revealing a pervasive presence of GFP and β-Gal positive cells suggesting that these cells most likely originated by cell fusion [12]. Similar studies have involved the implantation of human tumor cells into mice. For example, when Jacobsen and colleagues implanted adenocarcinoma cells from a pleural effusion of a female breast cancer patient into the mammary glands of nude mice, they found that the tumor was comprised of human and mouse cells [28]. In a cell line derived from one of the xenografts, approximately 30% of the mitotic cells had mixed mouse and human chromosomes, among which 8% carried mouse/human translocations [28]. However, because the patient-derived xenograft (PDX) cells that were injected were unlabeled and nontransgenic nude mice were used in this study, the TN-hybrid cells were not directly identified using fluorescence microscopy or flow cytometry, but rather by FISH analysis [28]. Genetically modified tumor cells expressing fluorescent reporters such as GFP, RFP or YFP were used in two other studies to investigate whether tumor cells would fuse with normal cells in vivo [20,29]. For example, Ramakrishnan et al. used GFP and RFP expressing ID8 ovary carcinoma cells to demonstrate that these cells do fuse with hematopoietic cells [20,29]. Both, injection of ID8-GFP cells into normal mice and injection of ID8-RFP cells into GFP mice resulted in TN-hybrid cells that were positive for either GFP and CD45 or GFP, RFP, and CD45, respectively [20]. More complex tumorigenic studies were recently performed by Gast and colleagues [29]. Here either H2B-RFP B16F10 mouse melanoma cells were injected into a GFP mouse, H2B-RFP/Cre B16F10 cells were injected into a R26R-stop-YFP transgenic mouse, or fl-dsRED-fl-GFP B16F10 cells were injected into a Cre mouse [29]. In all cases the authors could identify TN-hybrid cells, indicating that tumor cells could fuse with normal cells thereby giving rise to stable TN-hybrid cells.

Of course, selecting the perfect markers for the identification of TN-hybrid cells in in vitro and in vivo animal studies is simplified due to the wide range of genetically modified tumor cells and transgenic mouse strains available. However, this system is limited when studying cell fusion in a human cancer context. Current strategies to identify TN-hybrid cells in human cancers usually use marker molecules of the hybridization partner that are not commonly expressed by cancer cells. Because macrophages are well-known to be fusogenic [3] and several studies showed that macrophages could fuse with cancer cells in in vitro and in vivo animal studies [9,10,12,15], macrophage markers, such as CD14, CD45 and CD163, have primarily been used for the search of TN-hybrid cells in human tumors [13,20,29,35,36,41,42]. Shabo et al. found that CD163 expression in both breast and colorectal cancer samples was associated with metastatic spreading, early recurrence, and an overall poor prognosis [35,36,37]. However, in these studies, breast and colorectal cancer samples were not costained with cancer-specific markers, but rather were distinguished morphologically from tumor-associated macrophages. The pan-hematopoietic marker CD45 was applied to detect TN-hybrid cells in the ascites fluid of ovarian cancer patients and in the circulation of melanoma and pancreatic cancer patients [20,29,41,42]. Additionally, samples were counterstained with cancer-specific markers, such as cytokeratin or epithelial cell-adhesion molecule (EpCAM), to further support that the CD45 positive cells were most likely real TN-hybrids. The advantage of costaining procedures is the greater availability of appropriate negative controls. Only TN-hybrid cells are costained with, e.g., CD45 and EpCAM, whereas macrophages and tumor cells are single positive for CD45 or EpCAM, respectively.

While these studies show that putative TN-hybrid cells can be identified in human cancer patients based on the coexpression of hematopoietic markers and tumor-specific markers [20,29,41,42], a degree of uncertainty remains whether hematopoietic lineage markers are legitimate fusion markers, even though these findings are supported by appropriate animal studies [12,20,29]. Is the expression of, e.g., hematopoietic markers by tumor cells truly attributed to a former fusion event with cells of the hematopoietic lineage or rather, is it attributed to the tumor cells overall increased genomic instability? More reliable cell fusion data were obtained from human cancer patients who received a bone marrow transplant (BMT) [29,32,33,34,39]. In accordance with the abovementioned sex mismatch transplantation assays for stem cell-based regeneration studies, tumor samples of such cancer patients were assayed for both recipient and donor-specific DNA in the same samples. Donor DNA was clearly found in microdissected tumor samples obtained from a child that developed renal cell carcinoma and then metastases after BMT [32]. However, whether this finding is proof of cell fusion events is still not clear because A and O blood group alleles were used for determination. The recipient was O/O, the BMT donor was A/O, and the donor A allele was present throughout the metastasis, whereas the tumor genotype was A/O [32]. Since the source of the O allele in the tumor cells could not be determined it could have originated from both the donor and recipient [32]. Hence, it could not be ruled out that donor-derived cells may have first transdifferentiated into functional kidney cells and then underwent malignant transformation. In a similar study, a primary renal cell carcinoma of a female patient who received a BMT from her 15-year-old son was probed for the presence of the Y chromosome by FISH analysis [34]. Analysis of formalin-fixed histology specimens for cells containing the Y chromosome as a BMT cell marker and three or more chromosome 17s as a tumor cell marker showed that such cells were found in a region covering approximately 10% of the tumor area, located adjacent to normal renal tissue, where approximately 1% of the cells contained both markers [34].

Gast and colleagues also investigated female patients who received a sex-mismatched BMT, and that developed a pancreatic ductal adenocarcinoma [29]. Here, the Y chromosome was detected in approximately 4.3% of the cancer cells, which were counterstained with cytokeratin [29]. In addition, Y chromosome and cytokeratin positive tumor cells were also found in biopsies of solid tumors from women with previous sex-mismatched BMTs and renal cell carcinoma, head and neck squamous carcinoma (HNSCC) or lung carcinoma [29]. Moreover, Y chromosome positive and CD45 and EpCAM positive putative TN-hybrid cells (as well as CD45 and cytokeratin positive TN-hybrid cells) were also found in the circulation of female pancreatic cancer patients with metastatic disease and high levels of these cells were correlated with a poor prognosis [29]. However, metastases were not analyzed in this study, and because of that, it remains unclear whether circulating TN-hybrids truly exhibited an increased metastatic capacity. Nonetheless, these findings indicate a possible relationship between higher numbers of circulating TN-hybrid cells and the occurrence of distant metastases.

The above studies support the hypothesis that cell fusion events indeed occur in human cancers and that TN-hybrid cells can contribute to tumor progression and the formation of metastatic tumors. However, in all these studies only one donor marker was used, the Y chromosome. On the one hand, Y chromosome positive cancer cells indicate that they most likely stemmed from hematopoietic cells of the bone marrow. An alternative explanation for the presence of Y chromosome positive cancer cells could be due to fetal cell microchimerism (FCM), i.e., persistence of fetal cells in the mother [64]. FCM has been observed in a variety of human cancers, including breast, skin, lung, and cervix, whereby data are contradictory because some studies suggested a protective role and others a rather carcinogenic role for FCM [64]. Hence, to rule out the possibility of FCM, it has to be clarified whether sex-mismatch BMT female cancer patients had a male child.

A more sophisticated method to search for TN-hybrid cells in tumor samples of sex-mismatched BMT female cancer patients is the short tandem repeat (STR) analysis. The advantage of this technique, which is commonly used in forensic science and in paternity tests, is that TN-hybrid cells can be unequivocally identified via parallel determination of parental alleles located on different chromosomes. By doing so, Lazova et al. and LaBerge et al. were able to demonstrate an overlay of various donor and recipient alleles in microdissected tumor cells of male melanoma patients who received a BMT from their brothers [33,39]. To date, this is the most reliable proof that cell fusion events truly occur in human cancers.

These studies show that it is feasible to detect putative TN-hybrid cells in human cancers, but that it is much more difficult than in animal studies using genetically modified tumor cells and defined mouse strains. Only in BMT cancer patients could TN-hybrid cells be properly identified, but the number of BMT cancer patients is rather low compared to normal cancer patients. Coexpression of cancer-related markers, such as cytokeratins or EpCAM, and hematopoietic lineage markers, such as CD45 and CD163, is another strategy to identify putative TN-hybrid cells in human cancers. 

However, it has to be kept in mind that the current search strategies for TN-hybrid cells in human cancers are limited to those markers that are usually not expressed/present in tumor cells, such as the Y chromosome and hematopoietic lineage markers. Moreover, these search strategies also depend on the fact that these markers are stably expressed by the hybrid cells over time. However, what if TN-hybrid cells originate that do not express these specific markers or have lost them? This brings us to the next section.

## 3. Invisible or Dark Matter Hybrid Cells

To detect TN-hybrid cells in human cancers they must be “visible”, which means that they must express specific markers by which they can be detected. The studies summarized above likely indicate that tumor cells preferentially fuse with macrophages and that evolving TN-hybrids express macrophage markers. However, cell fusion is not limited to tumor cells and macrophages. It is well known that tumor cells also fuse with other tumor cells [47,53,65,66,67], as well as with fibroblasts [8,25,68] and stem(-like) cells [17,21,48,56,69,70]. Whether such tumor cell × tumor cell (TT)-hybrids or tumor cell × fibroblast/stem-like cell hybrid cells (also called TN-hybrid cells) can be detected will depend on whether they express specific markers. 

In this context, we would like to name such TT- and TN-hybrid cells that cannot be detected due to the lack of specific markers as “invisible” or “dark matter hybrids”. Dark matter is a hypothetical form of matter that constitutes 85% of all the matter in the Universe, but what it is made of is unknown [71]. Because dark matter does not interact with observable electromagnetic radiation, such as light, it is invisible and can only be detected indirectly due to interactions with gravitational forces [72]. In biology, the term dark matter has been suggested for species, such as intrinsically disordered proteins, posttranslational states, ion species, and rare, transient, and weak interactions undetectable by biochemical assays [73]. Like dark matter in gravitational physics, dark matter in biology is interacting and performing functions that are perceptible, yet cannot be directly detected or the matter itself cannot be sensed [73]. Noncoding DNA has been suggested as the genetic equivalent to dark matter in cosmology [74]. It is known that noncoding DNA affects gene expression, yet it has been difficult to determine its full impact due to the computationally intensive calculations needed to simultaneously process genomic and RNA expression data [74]. 

Dark matter hybrids could be the invisible part of the visible tumor matter. Hidden inside the tumor mass, these hybrids could interact and perform functions that contribute to tumor progression. Dark matter hybrids could originate from homotypic tumor cell fusion events. TT-hybrids may be phenotypically similar to parental nonfused tumor cells and cannot be discriminated from them (Figure 1A). However, dark matter hybrids could also originate from cell fusion events between tumor cells and normal cells lacking a suitable discriminatory marker expression pattern. Like TT-hybrids, such TN-hybrids would be indistinguishable from nonfused tumor cells (Figure 1B). Finally, dark matter hybrids could also originate from visible hybrid cells that have lost specific marker expression (Figure 1C). That TN-hybrid cells could lose marker expression over time has been recently demonstrated by Powell and colleagues [12]. Here, isolated and cultivated murine intestinal × macrophage hybrids lost the ability to express the murine macrophage marker F4/80 at the protein level [12]. Even though such TN-hybrid cells retained F4/80 expression at the mRNA level [12], they would be “invisible” by immunocytochemistry. In this context, it cannot be ruled out that even in sex-mismatched BMT human cancer studies former TN-hybrid cells may have lost the Y chromosome and become invisible.

The reason why TN-hybrid cells may lose marker expression and even whole chromosomes over time, and, hence, become part of the “dark matter” inside the tumor, is due to the increased genomic instability of the hybrid cells [75,76,77]. This is well known from hybridoma research [78] and the main cause for why hybridomas stop producing antibodies. Genomic instability is a hallmark of most, if not all, tumors and the main cause for the intratumoral heterogeneity [43]. Cell fusion is an inducer of genomic instability, which is related to the so-called heterokaryon to synkaryon transition (HST) process, which describes the fusion of the parental nuclei and the merging of the parental chromosomes. This process is accompanied by chromosomal rearrangements (e.g., translocations, deletions, etc.) and damage (single and double strand breaks), loss of whole chromosomes, unequal segregation of chromosomes in daughter cells and potentially even chromothripsis [29,48,75,76,77,79]. As an inducer of genomic instability, this is similar to the mutator phenotype that has been postulated for both the mutation and aneuploid theory of carcinogenesis and tumor progression [80,81]. However, if either mutations, aneuploidy or cell fusion can induce genomic instability, then the following question remains.

### Is it Possible to Distinguish between Mutation-Derived Genomic Instability, Aneuploidy-Derived Genomic Instability and Cell Fusion-Derived Genomic Instability?

Genomic instability is a hallmark of cancers and the main cause for the intratumoral heterogeneity of tumors [43]. All three carcinogenesis hypotheses (mutation, aneuploidy, and cell fusion) involve different mechanisms to explain this phenomenon. The mutation theory assumes that the malignant transformation of cells is attributed to a successive accumulation of driver and passenger mutations [82]. Passenger mutations could occur in any gene, but (most likely) do not have any impact on carcinogenesis, whereas driver mutations affect proto-oncogenes and tumor suppressor genes, thereby altering their functions in a way that ultimately could result in the malignant transformation of a cell [82]. In fact, mutations in genes that are involved in DNA mismatch repair and DNA proofreading are correlated with an increased rate of single nucleotide changes and chromosomal rearrangements [83,84] and could induce a so-called mutator phenotype [81].

In contrast, the aneuploidy hypothesis postulates that both the malignant transformation of a cell and genomic instability are attributed to an unequal and abnormal number of chromosomes [80,85,86,87]. This hypothesis was initially proposed by the German physician David von Hansemann and further developed by the German biologist Theodor Boveri more than 100 years ago [88,89]. They argued that the homeostasis of a cell is imbalanced, which would affect several cellular functions. Because an aneuploid tumor cell will always give rise to aneuploid daughter tumor cells, Li et al. proposed the term “autocatalytic karyotype evolution”, which means that the level of aneuploidy would increase with the number of cell divisions, thereby generating new lethal, preneoplastic and eventually neoplastic karyotypes [87]. Mutations in spindle checkpoint proteins and in proteins involved in chromosome segregation, as well as nonmutagenic carcinogens, were able to induce aneuploidy in cells and this was correlated with neoplastic transformation, genomic instability, and increased tumorigenicity [90,91,92,93,94]. 

As mentioned above, cell fusion-induced genomic instability is attributed to the HST process, which is the fusion of the parental nuclei and the merging of the parental chromosomes [48,76,77]. Why some TN-hybrid cells can undergo HST and how this process is regulated remains to be elucidated. It is known from stem cell-based tissue regeneration studies that heterokaryons derived from fusion events between hepatic cells and BMDCs can undergo ploidy reductions to generate daughter cells with one-half chromosomal content [95,96]. Interestingly, a more detailed analysis revealed that this process was not tightly regulated in individual cells and resulted in a variety of successful and failed bipolar, tripolar, and double mitoses [96]. In this context, successful means that, e.g., a double mitosis of 8n cells gave rise to four 2n daughter cells, whereas failed double mitosis resulted in the formation of two heterokaryons with two 2n nuclei [96].

Moreover, ploidy reductions were also associated with the unequal segregation of chromosomes, resulting in gains and losses of whole chromosomes and thereby giving rise to aneuploid karyotypes [96]. Whether ploidy reductions and HST are identical processes or if they differ remains to be elucidated. Nonetheless, both processes may have in common that the proliferation of the cells and the resolution of nuclear membranes is a prerequisite [48,76,96]. Otherwise, parental chromosomes could not be merged and segregated randomly to the daughter cells. The reason why both HST and ploidy reductions are associated with the missegregation of chromosomes is most likely attributed to extra centrosomes concomitant with multipolar spindles that would be sufficient to promote chromosome missegregation during bipolar and multipolar cell division [97]. Extra centrosomes in cells can originate by several mechanisms, such as overduplication, de novo synthesis of centrosomes, mitotic slippage, cytokinesis failure, and cell fusion [98]. In this context, it has to be considered that the process of HST/ploidy reductions will occur randomly in each hybrid cell. Thus, it cannot be predicted how cells will divide (i.e., bipolar mitosis, tripolar mitosis, multipolar mitosis or double mitosis), how chromosomes will be spread to the daughter cells (equal or unequal segregation), and whether chromosomes will be lost or not. All of this together potently drives aneuploidy and genomic instability in evolving TN-hybrid cells and more importantly, in their emerging progenies. The variation in chromosomal number between individual hybrid cell clones and in individual hybrid cell clones over time was recently shown by Zhou and coworkers [76]. Fusion-derived clones were near diploid at early passage and generally remained so during 10 to 11 passages, whereas the number of chromosomes in fusion-derived clones were near triploid or tetraploid at early passage and usually decreased with repeated passages, becoming near diploid [76]. 

In addition to the cell fusion induced aneuploidy and genomic instability in individual hybrid clones, Zhou et al. also observed an increased frequency of DNA damage, i.e., double-stranded breaks and translocations in hybrid cells [76]. Importantly, DNA damage is also related to chromosome missegregation during the division of aneuploid cells [99,100]. Moreover, chromosome missegregation and DNA damage both play key roles in the phenomenon known as chromothripsis, which is a catastrophic event in which one or a few chromosomes are shattered into tens to hundreds of fragments that are reassembled in random order; this gives rise to derivative chromosomes with extensive rearrangements, or alternatively the chromosomes may become lost and/or self-ligate into circular DNA structures called double minutes [101]. Hence, cell fusion, like mutations and aneuploidy, is a potent inducer of genomic instability. 

The question of this chapter was whether it would be possible to distinguish between mutation-derived genomic instability, aneuploidy-derived genomic instability, and cell fusion-derived genomic instability and the answer is clearly, no. Mutations, aneuploidy, and cell fusion are all potent inducers of genomic instability, and they collectively contribute to tumor progression. Mutations can result in aneuploidy and vice versa. Cell fusion also results in aneuploidy and, hence, mutations. It is simply not known what has happened in the evolution of the cancer cells found in the present tumor biopsy (Figure 2). 

As mentioned above, hematopoietic lineage markers or the Y chromosome are commonly used to detect putative TN-hybrid cells in human tumors. However, these markers are only as good as long as they are stable. Cell fusion is a potent inducer of genomic instability, and because of that, it cannot be ruled out that certain TN-hybrid cells could lose these markers over time. While such cells originated by cell fusion, they are now part of the dark matter (Figure 2). In this context, the following statement has to be considered.

## 4. Cell Fusion Is a 4D Process

Cell fusion may not be a single event that occurs only once in a tumor, but rather a repeating process occurring as the tumor develops. This is attributed to the chronically inflamed tumor microenvironment [102] and the finding that cell fusion is potently triggered by chronic inflammation and inflammatory cytokines, such as TNF-α, as a consequence of tissue injury [19,54,60,103,104,105,106,107,108]. Cell fusion is one mechanism by which BMDCs or myelomonocytic cells can restore organ tissue function [55,57,58,109,110]. However, those cells do not distinguish between “good” normal cells and “bad” neoplastic cells. As a consequence, the same cells that, e.g., restore liver function by cell fusion, will also restore tumor function by cell fusion. In addition to time (the longer a tumor exists the more cell fusion events will occur), the frequency of cell fusion events also depends on the size of a tumor (the bigger a tumor is the more cell fusion events will occur). Thus, it can be concluded that the number of TN-hybrid cells will steadily increase in the three-dimensional (3D) tumor environment with time (the fourth dimension) and because of that cell fusion is a 4D process (Figure 2). This, however, implies that more viable than nonviable TN-hybrid cells will originate by cell fusion during tumor development. 

The cell fusion in cancer hypothesis postulates that TN-hybrid cells can exhibit novel properties, such as enhanced drug resistance or enhanced metastatic capacity [30,79,111,112]. Because cell fusion is a 4D process, this implies that metastatic TN-hybrid cells can originate at any time in a tumor. Early dissemination of tumor cells has been observed for breast cancer, pancreatic cancer and malignant melanoma [113,114,115,116], indicating that the process of metastatic spreading has already started even before the primary tumor has been diagnosed. However, it cannot be ruled out that early evolved TN-hybrid cells are nonmetastatic at first but become metastatic later due to further genomic alterations. This would be similar to the postulated autocatalytic karyotype evolution hypothesis in which the degree of aneuploidy and genomic instability increases with time, thereby giving rise to invasive and metastatic karyotypes [87]. Nonmetastatic TN-hybrid cells could be positive for fusion marker expression and could retain marker expression over time despite genomic instability. However, genomic instability could also result in a loss of fusion marker expression, and such cells would become part of the dark matter (Figure 2). 

In summary, cell fusion could be a 4D process resulting in a steady increase in TN-hybrid cells over time within the tumor tissue with the possibility that TN-hybrid cells can evolve at any time. However, the following question still remains unclear.

## 5. How Many TN-Hybrid Cells are Needed for Tumor Progression?

In fact, there is no answer to this question. Cancer is a highly individual disease. Even though two patients might have the same type of cancer, the outcome could be totally different. Patient A may respond to therapy and be cured, while patient B may not. Likewise, patient B may not develop metastases despite a large primary tumor, whereas during primary tumor formation in patient A, cancer cells may have already disseminated. 

Cell fusion has been associated with TN-hybrid cells that exhibit novel properties, such as an increased metastatic capacity or an enhanced drug resistance [30,79,111,112]. However, if a tumor has not metastasized or is sensitive to cancer therapy, this does not mean that TN-hybrid cells have not formed. Cell fusion is an open and random process that can also give rise to nonmetastatic and drug sensitive TN-hybrid cells. Such cells would rather contribute to the growth and heterogeneity of the primary tumor. On the other hand, several studies have demonstrated that the origin of TN-hybrid cells is associated with disease progression [7,8,9,10,11,12,13,14,15,16,17,18,19,20,21,22,23,24,25,26,27,28,29,30,31,32,33,34,35,36,37,38,39,40,41,42]. 

As summarized above, the proportion of TN-hybrid cells within analyzed human tumor samples varied markedly between different human cancer types. In some studies, the amount of TN-hybrid cells was approximately 20 to 40% (and even higher) [13,20,34,36,38], whereas in another study only approximately 4% of all cancer cells were thought to be TN-hybrid cells [29]. However, these numerical values only reflect those TN-hybrid cells that were clearly identified by marker expression. The number of TN-hybrid cells that belong to the dark matter inside the tumor remains unclear. In any case, these findings likely suggest that higher numbers rather than lower numbers of TN-hybrid cells are likely to be associated with tumor progression. For instance, both the overall survival and distant recurrence-free survival were significantly decreased in breast cancer patients with more than 25% of tumor cells CD163 positive [13]. Similar findings were reported for colorectal cancer patients demonstrating that CD163 expression in tumor cells was correlated with a significantly decreased cumulative survival and local recurrence-free survival [36]. However, in a recently published study, Gast and colleagues showed that the number of TN-hybrid cells in pancreatic ductal adenocarcinoma was only approximately 4.3% [29]. Nonetheless, significantly higher numbers of circulating CK+/CD45+ TN-hybrid cells were found in patients with distant metastases [29]. Likewise, high numbers of circulating CK+/CD45+ TN-hybrid cells were further correlated with significantly decreased overall survival compared to patients with low numbers of circulating CK+/CD45+ TN-hybrid cells [29]. In contrast, no correlation with stage or survival was observed in patients with conventionally circulating tumor cells [29]. Even though it remains unknown whether circulating CK+/CD45+ TN-hybrid cells truly exhibited an increased metastatic capacity since metastases were not further analyzed, these findings may indicate that even lower numbers of TN-hybrid cells in the primary tumor might contribute to tumor progression. This would also apply to the abovementioned data on early dissemination [113,114,115,116] if such metastatic cells have originated by cell fusion. 

## 6. Conclusions

In conclusion, an increasing body of evidence indicates that cell fusion events in human cancers are a real phenomenon and, hence, it can be concluded that TN-hybrid cells do truly contribute to tumor progression by increasing tumor heterogeneity and most likely by triggering metastasis. As summarized above, the expression of hematopoietic markers or the presence of donor DNA in tumor cells are current and suitable strategies to identify TN-hybrid cells. However, these strategies require that TN-hybrid cells retain these markers over time. A loss of these markers is equal to a loss of visibility and such TN-hybrid cells become part of the dark matter inside the tumor tissue. The probability that TN-hybrid cells may lose these markers over time is due to their genomic instability. Likewise, dark matter TN-hybrid cells may originate from tumor cells and normal cells that express similar markers or alternatively the dark matter TN-hybrid cells may have adopted the phenotype of their normal cell fusion partner, thereby mimicking harmless cells.

These considerations indicate that the cell fusion in cancer hypothesis is much more complex than initially thought and much more work needs to be done. The first step demonstrating that cell fusion in human cancers is a real phenomenon has been accomplished. However, much more work is necessary. In addition to cell fusion, epithelial–mesenchymal plasticity (EMP), dedifferentiation, and trans differentiation can be viewed as the four major sources of tumor plasticity and heterogeneity during tumorigenesis, and they all share some common characteristics [117]. Thus, the challenge will be not only to identify new cell fusion specific markers to make dark matter TN-hybrid cells visible but also to show that these markers will be suitable in distinguishing dark matter hybrids from cancer cell plasticity caused by EMP or transdifferentiation. Whether genomic DNA, epigenetic alterations, specific mRNAs, or specific proteins could be such specific markers remains to be clarified. However, we recommend that a combination of different markers of both cell fusion partners will be necessary to clearly identify hybrid cells and to distinguish them from nonhybrid cells. 

Likewise, it has to be clarified whether human TN-hybrid cells truly exhibit an increased metastatic capacity and an enhanced drug resistance as postulated by the hypothesis. Initial data are promising [7,29,33,34,39,41], but need to be validated in future studies. For instance, it would be of interest to analyze metastases for the presence of hybrid cells. This has only been done in a few studies so far [32,33,39] but is mandatory for proving the hypothesis. Likewise, tumor relapses should be analyzed for TN-hybrids. If cell fusion can truly give rise to drug-resistant hybrid cells, this process may play a role during chemotherapy or radiation therapy. Such therapies are effective at eradicating cancer cells but will also induce a wound healing response. If BMDCs or myelomonocytic cells or other to date unknown cells can restore cancer therapy damaged tumor cells by cell fusion, this could result in cancer therapy resistant TN-hybrid cells [16,79].

Furthermore, the mechanism of cell fusion has to be resolved. Cell fusion is a rather a tightly regulated process that has to be both initiated and terminated. Even though cell fusion plays a crucial role in physiological and pathophysiological processes, considerably less is still unknown about this phenomenon. Inflammation is a positive trigger for cell fusion, but what happens to the cells under inflammatory conditions? Which proteins are upregulated and which are downregulated in the cells? Do cells randomly fuse with each other (e.g., macrophages fuse with any tumor cells they are in contact with) or is there a directed fusion (e.g., macrophages will only fuse with certain tumor cells, which possibly express proteins marking them as fusion partners)? This also applies to the question of why the proportion of TN-hybrid cells in human tumors varies greatly between different types of cancer? Are certain types of cancer preferred for cell fusion because they likely express cell fusion related proteins? 

Thus, there are many questions that have to be answered and much work that has to be done in the future to shed more light on the dark matter hypothesis of cell fusion. 

## Figures and Tables

**Figure 1 cells-08-00132-f001:**
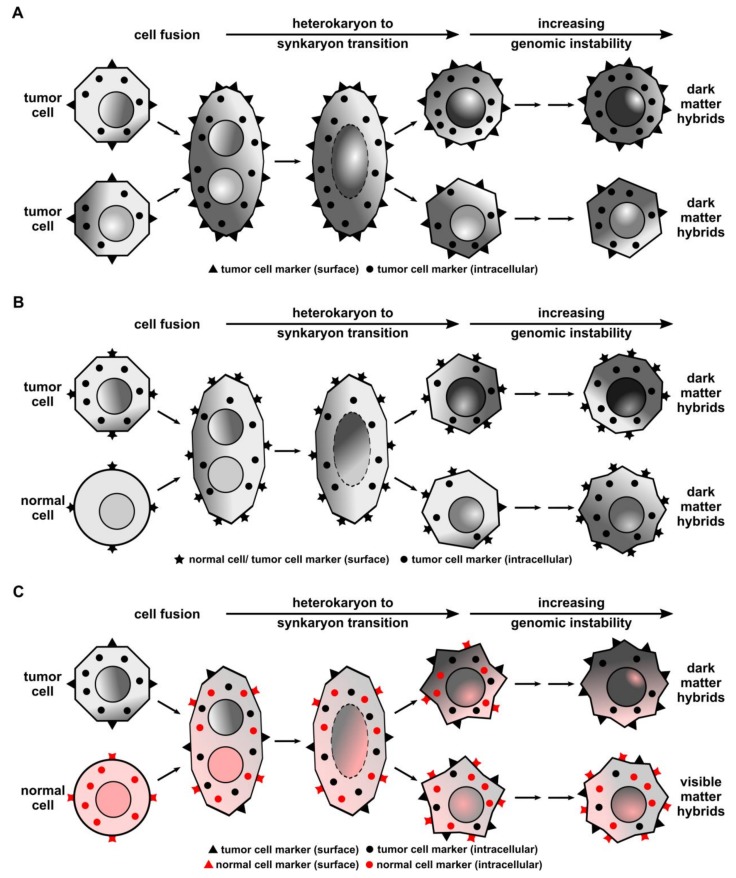
Model of how dark matter hybrids may originate. Tumor cells can either fuse with other tumor cells (homotypic fusion) or normal cells (heterotypic fusion), resulting in a heterokaryon (a hybrid cell with at least two nuclei). Hybrid cells can undergo a process known as the heterokaryon-to-synkaryon transition (HST), which is the merging of the parental chromosomes and random distribution to (at least) two daughter cells with one nucleus (synkaryon). A solid-colored nucleus represents the unchanged karyotype of a normal cell, whereas the altered karyotype of a tumor cell is presented as a gradient colored nucleus. HST is a potent inducer of genomic instability and most hybrid cells will die or will be less capable of proliferation (not shown here). (**A**) Homotypic tumor cell fusion results in dark matter hybrids that are indistinguishable from parental cells. (**B**) Heterotypic fusion of a tumor cell and a normal cell both exhibiting a similar specific marker pattern also results in dark matter hybrids that are indistinguishable from parental cells. (**C**) Heterotypic fusion of a tumor cell and a normal cell exhibiting a specific discrimination marker pattern. Due to genomic instability one TN-hybrid cell loses discrimination markers over time and becomes a dark matter hybrid. In contrast, the other TN-hybrid cell retains discrimination markers and becomes part of the visible matter hybrids.

**Figure 2 cells-08-00132-f002:**
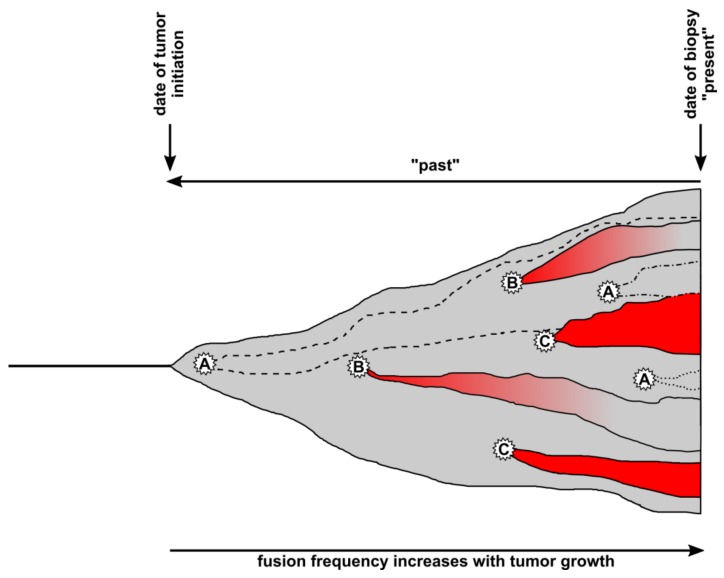
Timeline of cell fusion events inside a tumor tissue. Cell fusion events can occur at any time inside a tumor tissue (gray), whereby the fusion frequency should increase with tumor size. Whether hybrid cells will become visible or part of the dark matter (invisible) will depend on which cell types fuse and if they retain or lose specific marker expression. A, Tumor cells could either fuse with each other (homotypic fusion) or with normal cells expressing similar markers (heterotypic fusion) resulting in invisible/dark matter hybrids. Such hybrids are indistinguishable from nonfused tumor cells (gray) but contribute to tumor heterogeneity and tumor growth (dashed and dotted lines). B, Heterotypic cell fusion between tumor cells and normal cells results in TN-hybrid cells that have lost specific marker expression over time and become part of the dark matter. A switch from red to gray indicates the loss of specific marker expression. C, Heterotypic cell fusion between tumor cells and normal cells results in TN-hybrid cells that have retained normal cell marker expression over time and can be detected in the biopsy.

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
