# Peer review of "Cell Fusion in Human Cancer: The Dark Matter Hypothesis"

_cells, 2019, doi:10.3390/cells8020132_

Round 1
Reviewer 1 Report
See attached file.

Author Response
Reviewer #1
We would like to thank the reviewer for his/her helpful comments. In the following we will reply to the comments in a point-to-point manner.
In this manuscript, Weiler and Dittmar review evidence fusion of tumor cells with normal cells may contribute to progression of cancer and summarize some mechanisms that potential explain why cell fusion is not more evident in a given tumor or in tumors generally. Because these mechanisms make it difficult and in some instances impossible to detect the progeny of fused cells, the authors refer to the progeny as "dark matter." The senior author, Dr. Dittmar, has published much on the subject of cell fusion in cancer - this manuscript is the best if his many contributions.
Although not discussed or even mentioned in the manuscript, the term "dark matter" is intended to bring to mind that large fraction of matter in the universe that is not directly observable but inferred from its impact on matter that is observable. This analogy is quite apt and the authors can best exploit it by stating at the beginning of the Introduction that the progeny of fusion of cancer cells and other cells might be discernable only indirectly, akin to "dark matter" of astrophysics.
We agree with the reviewer that this important information was missing in the previous version of the manuscript. Hence, we divided the former chapter 2.1 into two parts. In the new chapter 2.2 we will briefly summarize the topic dark matter. We initially thought to describe dark matter first in the introduction, but felt that it would be better summarized in an own chapter.
The review of clinical and some experimental evidence cell fusion occurs in some or all cancers is quite good and the assertions that (a) proof that a process like cell fusion can occur does not constitute proof that it does occur and (b) documentation that cell fusion does occur cannot be taken as proof that cell fusion has one or another impact are important and commendable cautions.
The figures and several sections of this otherwise excellent manuscript could be improved and/or clarified.
1. Figure 1 appears illustrates some mechanisms that obscure the hybrid origin of tumor cells. However, the processes are better explained in the text than in the figure and identifying what processes are "modeled" in the figure is far more difficult than understanding the processes. The authors could make the figure more useful by naming the process they are modeling in each of the vertical panels (also, readers generally find it easier to grasp series of steps that are depicted horizontally rather than vertically) and by showing or in some way listing alternative mechanisms through which the karyotype or phenotype of the final tumor cells could be generated (as the authors clearly state few would question whether cell fusion can generate the end mixture of cancer cells, the problem is that other mechanisms can generate the same mixture).
Panel 3A is evidently aimed at depicting how fusion of two cancer cells leading to tetraploidy generates chromosomal instability. The figure would be improved by inserting a fused cell with two nuclei before the mononuclear tetraploid cell is shown. More important, the figure does not convey the problem that chromosomal instability also arises with faulty mitosis. Also unclear is what the authors are trying to convey about tumor cell markers - do they want readers to imagine the figure to suggest that quantitative differences in chromosomal DNA are reflected in quantitative differences in expression of the same markers? The legend suggests the reader should compare markers with parental cell markers - is it the quantity that should be compared? Tumors containing cells with various ploidies and recombination and differences in the apparent level of expression of a tumor cell marker could reflect aberrant mitosis or fusion or both, but the figure only depicts how cell fusion generates the outcome.
Panel 3B appears to be aimed at modeling how tumor cells derived from fusion of normal cells with parental tumor cells can exhibit a phenotype or karyotype different from both of these. However, several aspects of the figure detract from clarity. First the normal cell contains nicely paired chromosomes but tumor cells in this and other panels contain few paired chromosomes--what are the authors trying to depict with that? Are the authors trying to say that tumor cells always have abnormal paring? If that is the case then one need not invoke cell fusion to explain aneuploidy, recombination, etc. Why does the normal cell have no markers (in contrast to panel 3C? The final product is similar to the final product in panel A but still more mechanisms are left out.
Panel 3C raises still more questions, the most important of which is why the presence of a hybrid cell is apparent in this panel but not in the other panels. The authors apparently want the reader to conclude that only in panel C is the hybrid origin apparent because the tumor begins to express a marker characteristic of normal cells. But, does such a marker exist? Does anyone believe that de novo expression of a "normal cell" marker in a tumor ipso facto proves that cell fusion occurred? Because the normal cell and tumor cell at the top have completely different markers, the figure might be illustrating the rare example of development or progression of a tumor in a chimeric individual - if that is the intent it should be explained. If that is not the intent then at the outset the tumor and normal cells should share at least one marker (as do karyotypically different cells in panels A and B.
The figure leaves out the most important mechanism underlying the dark matter phenomenon and that is the dilution of hybrid archetypes by the combination of proliferation and selection. This mechanism, illustrated to some extent in Figure 2, explains how diversification of sets of tumor cells fuels tumor evolution and it is evolution that hides the past history of cells. When hybridization causes aneuploidy, recombination and mutation, the most extreme and readily detected progeny die or are less capable of proliferation and hence the karyotype and genome evolve away from the synkaryon. I think the figure should be refashioned to emphasize the key mechanisms that are nicely explained in the text of the paper.
We have revised Figure 1 thoroughly. As suggested by the reviewer “heterokaryons” have been added to the revised figure and all steps are now depicted horizontally. Likewise, chromosomes were removed and replaced by either solid-colored nucleus (representing a normal karyotype of a normal cell) and gradient colored nucleus (representing an altered karyotype of a tumor cell).
However, the primary aim of Figure 1 was to summarize which cell types could fuse with each other and whether visible (detectable) or invisible (undetectable/ dark matter) hybrids will evolve. We thought about the possibility to add “cell death” to the figure, but we felt that this would make the figure to complex. Nevertheless, we added this important information to the figure legend.
Current strategies to detect tumor cell ´ normal cell hybrids depends on the proof of markers that are commonly not expressed by tumor cells, like hematopoietic markers or the Y-chromosome in cancer cells of female patients. Such hybrids will be visible since they can be identified by these markers. However, if such hybrids will lose such markers due to genomic instability they will switch from “visible” to “invisible”. Likewise, if two cells will fuse lacking appropriate markers, like two tumor cells or a tumor cell and a normal cell, which possess a similar surface expression pattern, invisible hybrid cells will also evolve.
The term “marker” was used for molecules that could be used for discrimination between tumor cells and normal cells and that can be used for identification of hybrid cells.
2. Figure 2 is better because it is simpler but it appears to misrepresent the authors' point. To the reader's right, the figure models a heterogeneous tumor, containing the progeny of parental cells and of hybrids formed at various times during the life of the tumor (toward the left). Since each cell fusion event is depicted as a different shade of red, it appears that progeny of events of "A" type and "C" type remain unchanged over time and are detected in the final tumor on the right. In contrast, fusion event type "B" continues to change over time so that the progeny on the right are indistinguishable from the parental fusion partners and the tetraploid hybrid. Fusion B does appear to model the dark matter phenomenon but fusions A and C do not. Yet, the figure appears to suggest that most fusion events can be detected in the final tumor, whereas I think the authors intend to portray how evidence of cell fusion disappears over time. I think the figure would be much better if all of the post-fusion bands changed color and darkness like "B" and the parental tumor cells remained a relatively stable light gray (as is shown). I also suggest that the authors explain what they mean by the width of the bands---I assume it is tumor mass and if that is so then they might also wish to "slow" the growth of the light gray band because it appears to exhibit log growth early and slow growth later. I suspect the authors would prefer to suggest that cell fusion can accelerate growth (progression).
The primary aim of Figure 2 was to visualize that i) different types of cell fusion events could occur in a tumor, which could give rise to “visible” or “invisible/dark matter” hybrids and that ii) the number of cell fusion events should increase with tumor size. We agree that in its present form this figure might be misinterpreted since type “A” hybrids, which were colored in different shades of gray could be clearly distinguished from the primary tumor mass that was colored in light gray. However, as suggested in this manuscript invisible/ dark matter hybrids should be rather indistinguishable from non-fused tumor cells.
Thus, Figure 2 was revised. Instead of using different shades of gray we now used different line types to illustrate that type “A” cell fusion events could give rise to “invisible/ dark matter” hybrids, whereby post-fusion bands were also colored in light gray. Likewise, post-fusion bands of type “B” cell fusion events were also changed from dark gray to light gray to show that hybrid cells can switch from “visible” to “invisible” if they have lost a specific visibility marker.
A few minor points are as follows:
At the outset the authors state - "The cell fusion in human cancer hypothesis remains a mystery"--what is the hypothesis? And, what aspect of that hypothesis is a mystery?
The abstract was revised and this sentence was removed.
On some occasions the authors use the term "genomic instability" when referring to expression of heterotypic markers on cells--i.e. that an unstable genome can allow any marker to be expressed on any type of cell. It would help readers if they explain this mechanism when first mentioned.
We have added a short explanation of genomic instability to the introduction section. Likewise, we mentioned that cell fusion is also an inducer of genomic instability, which would result in both viable and non-viable hybrid cells. Line 53-62: “…,which is a hallmark of most, if not all, tumors and the main cause for intratumoral heterogeneity [43]. Genomic instability generates new mutations and/or gross chromosomal aberrations in dividing tumor cells [44]. This can be beneficial for the overall capacity of a tumor to adapt changes in its environment [44]. However, newly acquired genetic alterations can also compromise the genetic dominance of the tumor cells and thus affect tumor cell viability [44]. In this context it should be noted that cell fusion is also a potent inducer of genomic instability. Hence, cell fusion can give rise to hybrids that may adapt better to changes in the tumor environment or to cancer therapy but can also give rise to nonviable hybrids. Likewise, hybrid cells may lose specific cell fusion markers over time as a result of genomic instability, thereby becoming indistinguishable from nonfused tumor cells.”
In "cell fusion is a 4D process" the authors propose a complex and important hypothesis---that the frequency of cell fusion in a cancer should increase as a log function of tumor mass (3-dimesions) and time. The authors might wish to acknowledge however that the impact of cell fusion both on the genesis and on the progression of cancer is conditioned by factors other than these 4 dimensions: (i) the contribution of cell fusion or any process to tumor evolution is conditioned by selection (e.g. as mentioned above, chromosomal changes reflecting fusion are lost over time as less fit cells are diluted); (ii) the hypothesis does not explain differences in the frequency and biology of cancers between various tissues; (iii) if one accepts the preeminence of the 4 dimensions, as the authors put them, the contribution of "4-D" to initiation of cancer (i.e. when only one or two dimensions exist) must be quite different than the contribution to progression. Perhaps a sentence to that point might be added.
The reviewer is right with this comment that the impact of cell fusion depend on more factors and processes than 4 dimensions. However, the intention of this paragraph and the 4D hypothesis was a different one. Of course, cell fusion is an open and random process. Hence, the ultimate phenotype of the evolving hybrids cannot be predicted, which also applies to the viability of the evolving hybrids. Most hybrid cells will likely die and only a few will survive. And those that will survive could be more malignant or not. Here, we fully agree with the reviewer.
But when do cell fusion events occur inside the tumor? Is cell fusion an early or a late event? Does it occurs only once at a certain time point and place inside the tumor or is a repeating event that is maintained by the chronically inflamed tumor microenvironment? And what is the frequency of cell fusion events inside a tumor? All these considerations led to the 4D hypothesis. The 4D concept is also important to understand that due to genomic instability TN-hybrids could change their phenotype and that it could take time till TN-hybrid cells became metastatic. We thus hope that the reviewer agrees that we do not have revised this section. In any case, we added the following sentence to the manuscript (lines 369-370: “This, however, implies that more viable than non-viable TN-hybrid cells will originate by cell fusion during tumor development.”) to indicate that also non-viable hybrids can originate.
The manuscript suffers most from poor English usage and grammar. The errors are too numerous to list and I will list a few from the beginning:
(a) The abstract states: "However, such cell fusion search strategies are only as good as stable these cell fusion related markers are expressed/ being present in TN-hybrid-derived cells"???
(b) Even though it is well-known that the biological phenomenon plays a crucial role in several physiological processes, like fertilization, placentation, formation of myofibers, osteoclastogenesis and tissue regeneration/ wound healing (for review see: [1-4]) its role and impact in cancer is still controversially debated."
(c) This particularly applies to the question whether cell fusion events do really occur in human cancers and if, whether the evolving tumor cell x normal cell hybrids and their progenies (thereafter named TN-hybrid-derived cells) do truly contribute to disease progression as was proposed by the German physician Otto Aichel in 1911"
The authors should consider engaging a professional copyeditor to ensure corrects do not distort the meaning of their manuscript.
The manuscript was language edited by American Journal Experts.
Reviewer 2 Report
In this manuscript, Weiler and Dittmar review evidence fusion of tumor cells with normal cells may contribute to progression of cancer and summarize some mechanisms that potential explain why cell fusion is not more evident in a given tumor or in tumors generally. Because these mechanisms make it difficult and in some instances impossible to detect the progeny of fused cells, the authors refer to the progeny as "dark matter." The senior author, Dr. Dittmar, has published much on the subject of cell fusion in cancer - this manuscript is the best if his many contributions.
Although not discussed or even mentioned in the manuscript, the term "dark matter" is intended to bring to mind that large fraction of matter in the universe that is not directly observable but inferred from its impact on matter that is observable. This analogy is quite apt and the authors can best exploit it by stating at the beginning of the Introduction that the progeny of fusion of cancer cells and other cells might be discernable only indirectly, akin to "dark matter" of astrophysics.
The review of clinical and some experimental evidence cell fusion occurs in some or all cancers is quite good and the assertions that (a) proof that a process like cell fusion can occur does not constitute proof that it does occur and (b) documentation that cell fusion does occur cannot be taken as proof that cell fusion has one or another impact are important and commendable cautions.
The figures and several sections of this otherwise excellent manuscript could be improved and/or clarified.
1. Figure 1 appears illustrates some mechanisms that obscure the hybrid origin of tumor cells. However, the processes are better explained in the text than in the figure and identifying what processes are "modeled" in the figure is far more difficult than understanding the processes. The authors could make the figure more useful by naming the process they are modeling in each of the vertical panels (also, readers generally find it easier to grasp series of steps that are depicted horizontally rather than vertically) and by showing or in some way listing alternative mechanisms through which the karyotype or phenotype of the final tumor cells could be generated (as the authors clearly state few would question whether cell fusion can generate the end mixture of cancer cells, the problem is that other mechanisms can generate the same mixture).
Panel 3A is evidently aimed at depicting how fusion of two cancer cells leading to tetraploidy generates chromosomal instability. The figure would be improved by inserting a fused cell with two nuclei before the mononuclear tetraploid cell is shown. More important, the figure does not convey the problem that chromosomal instability also arises with faulty mitosis. Also unclear is what the authors are trying to convey about tumor cell markers - do they want readers to imagine the figure to suggest that quantitative differences in chromosomal DNA are reflected in quantitative differences in expression of the same markers? The legend suggests the reader should compare markers with parental cell markers - is it the quantity that should be compared? Tumors containing cells with various ploidies and recombination and differences in the apparent level of expression of a tumor cell marker could reflect aberrant mitosis or fusion or both, but the figure only depicts how cell fusion generates the outcome.
Panel 3B appears to be aimed at modeling how tumor cells derived from fusion of normal cells with parental tumor cells can exhibit a phenotype or karyotype different from both of these. However, several aspects of the figure detract from clarity. First the normal cell contains nicely paired chromosomes but tumor cells in this and other panels contain few paired chromosomes--what are the authors trying to depict with that? Are the authors trying to say that tumor cells always have abnormal paring? If that is the case then one need not invoke cell fusion to explain aneuploidy, recombination, etc. Why does the normal cell have no markers (in contrast to panel 3C? The final product is similar to the final product in panel A but still more mechanisms are left out.
Panel 3C raises still more questions, the most important of which is why the presence of a hybrid cell is apparent in this panel but not in the other panels. The authors apparently want the reader to conclude that only in panel C is the hybrid origin apparent because the tumor begins to express a marker characteristic of normal cells. But, does such a marker exist? Does anyone believe that de novo expression of a "normal cell" marker in a tumor ipso facto proves that cell fusion occurred? Because the normal cell and tumor cell at the top have completely different markers, the figure might be illustrating the rare example of development or progression of a tumor in a chimeric individual - if that is the intent it should be explained. If that is not the intent then at the outset the tumor and normal cells should share at least one marker (as do karyotypically different cells in panels A and B.
The figure leaves out the most important mechanism underlying the dark matter phenomenon and that is the dilution of hybrid archetypes by the combination of proliferation and selection. This mechanism, illustrated to some extent in Figure 2, explains how diversification of sets of tumor cells fuels tumor evolution and it is evolution that hides the past history of cells. When hybridization causes aneuploidy, recombination and mutation, the most extreme and readily detected progeny die or are less capable of proliferation and hence the karyotype and genome evolve away from the synkaryon. I think the figure should be refashioned to emphasize the key mechanisms that are nicely explained in the text of the paper.
2. Figure 2 is better because it is simpler but it appears to misrepresent the authors' point. To the reader's right, the figure models a heterogeneous tumor, containing the progeny of parental cells and of hybrids formed at various times during the life of the tumor (toward the left). Since each cell fusion event is depicted as a different shade of red, it appears that progeny of events of "A" type and "C" type remain unchanged over time and are detected in the final tumor on the right. In contrast, fusion event type "B" continues to change over time so that the progeny on the right are indistinguishable from the parental fusion partners and the tetraploid hybrid. Fusion B does appear to model the dark matter phenomenon but fusions A and C do not. Yet, the figure appears to suggest that most fusion events can be detected in the final tumor, whereas I think the authors intend to portray how evidence of cell fusion disappears over time. I think the figure would be much better if all of the post-fusion bands changed color and darkness like "B" and the parental tumor cells remained a relatively stable light gray (as is shown). I also suggest that the authors explain what they mean by the width of the bands---I assume it is tumor mass and if that is so then they might also wish to "slow" the growth of the light gray band because it appears to exhibit log growth early and slow growth later. I suspect the authors would prefer to suggest that cell fusion can accelerate growth (progression).
A few minor points are as follows:
At the outset the authors state - "The cell fusion in human cancer hypothesis remains a mystery"--what is the hypothesis? And, what aspect of that hypothesis is a mystery?
On some occasions the authors use the term "genomic instability" when referring to expression of heterotypic markers on cells--i.e. that an unstable genome can allow any marker to be expressed on any type of cell. It would help readers if they explain this mechanism when first mentioned.
In "cell fusion is a 4D process" the authors propose a complex and important hypothesis---that the frequency of cell fusion in a cancer should increase as a log function of tumor mass (3-dimesions) and time. The authors might wish to acknowledge however that the impact of cell fusion both on the genesis and on the progression of cancer is conditioned by factors other than these 4 dimensions: (i) the contribution of cell fusion or any process to tumor evolution is conditioned by selection (e.g. as mentioned above, chromosomal changes reflecting fusion are lost over time as less fit cells are diluted); (ii) the hypothesis does not explain differences in the frequency and biology of cancers between various tissues; (iii) if one accepts the preeminence of the 4 dimensions, as the authors put them, the contribution of "4-D" to initiation of cancer (i.e. when only one or two dimensions exist) must be quite different than the contribution to progression. Perhaps a sentence to that point might be added.
The manuscript suffers most from poor English usage and grammar. The errors are too numerous to list and I will list a few from the beginning:
(a) The abstract states: "However, such cell fusion search strategies are only as good as stable these cell fusion related markers are expressed/ being present in TN-hybrid-derived cells"???
(b) Even though it is well-known that the biological phenomenon plays a crucial role in several physiological processes, like fertilization, placentation, formation of myofibers, osteoclastogenesis and tissue regeneration/ wound healing (for review see: [1-4]) its role and impact in cancer is still controversially debated."
(c) This particularly applies to the question whether cell fusion events do really occur in human cancers and if, whether the evolving tumor cell x normal cell hybrids and their progenies (thereafter named TN-hybrid-derived cells) do truly contribute to disease progression as was proposed by the German physician Otto Aichel in 1911"
The authors should consider engaging a professional copyeditor to ensure corrects do not distort the meaning of their manuscript.
Author Response
Reviewer #2
We would like to thank the reviewer for finding the time to critically read the manuscript and for giving us helpful comments. In the following we will reply to his/her comments.
This is an excellent review on cancer cell fusion, which is a very important but long-overlooked mechanism of cancer progression and metastasis. Cancer cell fusion as a mechanism of cancer progression is intentionally overlooked by cancer research community, because the current cancer research is dominated by mutation theory. Most researchers choose gene mutation as their research subject, since cancer cell fusion is too difficult to investigate. In this sense, this reviewer strongly suggests publication of this review to attract research interests in cancer cell fusion. On the other hand, this reviewer thinks that current version of the manuscript should be revised to meet the standard of journal publication.
Major comments:
1. On dark matter “phenomenon”.
Dark matter, defined as “a hypothetic form of matter”, is hypothetical and could not be proved by conventional methodology. In this review, dark matter refers to tumor × normal hybrid derived cells, which do exist but are difficult to detect with current cancer research methods.
The authors should decide whether the term “dark matter” is appropriate to be used to describe tumor × normal hybrid derived cells. Should the authors not be confident on the existence of this type of cells, “dark matter hypothesis” should be used. Dark matter per se is hypothetical, and it will never be a phenomenon.
2. On the introduction of dark matter.
The term of dark matter is being used widely in cancer research, referring to various unknowns (e.g., effects of chaos in biomedical alterations, or possible function of non-coding DNA in the genome, etc.). To help readers, the term of “dark matter” in this review should be clearly defined at the beginning of the manuscript. The Abstract introduces this term only with “...the pool of invisible or so-called dark matter TN-hybrid-derived cells inside a tumor…”, but failed to point out who called these cells dark matter.
We will respond to both points with a common response. We agree with the reviewer that in the previous version of the manuscript important informations about “dark matter” were missing. We decided to introduce the topic dark matter in an own chapter. Therefore, chapter 2.1 was divided into two chapters, whereby the new chapter 2.2 was titled “Invisible or dark matter hybrid cells. Here, we will briefly summarize the topic “dark matter”. We initially thought to describe dark matter in the introduction, but felt that it would be better summarized in an own chapter.
Likewise, the title was changed and “phenomenon” was replaced by “hypothesis”.
We have also rewritten the abstract.
3. On detection of tumor × normal hybrid derived cells.
According to this review, dark matter tumor × normal hybrid derived cells do exist but are difficult to detect. This reviewer thinks that the review should be more critical in reviewing the published literature. It would be great if the manuscript could change the title of Section 3.1 (into What are the current methods for detecting tumor × normal hybrid derived cells?), and then add a separate section at the end of the review to discuss the following issues:
Of course, different methods have been used to detect TN-hybrid cells, but ultimately TN-hybrid cells have been identified and distinguished from non-hybridized tumor cells by specific or ideal markers. Therefore, we do not want to change the title of chapter 2.1.
1) What is the flaw of current detection of tumor × normal hybrid derived cells?
This is an interesting question and we think that we have discussed this thoroughly in chapter 2.1.
2) Should gene expression or protein marker be used as proof of tumor × normal hybrid derived cells?
3) How to distinguish tumor × normal hybrid derived cells from cancer cell lineage plasticity caused by transdifferentiation, cancer stem cell property, epithelial to mesenchymal transition, mesenchymal to epithelial transition, and tissue cell mimicry?
We would like to thank the reviewer for these interesting and important questions that have been added to the conclusion section
4) Theoretically, what is the definitive proof of tumor × normal hybrid derived cells? Considering that most cancer patients do not have a transplant history, how the detection of tumor × normal hybrid derived cells could ever be translated into clinical cancer detection and prognosis?
Once we know what the definitive proof of TN-hybrid cells will be we will be able to answer this question. But as long as we do not know this we will not be able to answer this question. We apologize, but we have not added this theoretical question to the manuscript.
Minor comments:
1. The manuscript should be critically edited for the use of standard English.
The manuscript was language edited by American Journal Experts.
2. Enumeration of the sections should be revised.
Enumeration of the sections has been revised.
Reviewer 3 Report
Julian Weiler and Thomas Dittmar
Cell fusion in human cancer: the dark matter phenomenon.
This is an excellent review on cancer cell fusion, which is a very important but long-overlooked mechanism of cancer progression and metastasis. Cancer cell fusion as a mechanism of cancer progression is intentionally overlooked by cancer research community, because the current cancer research is dominated by mutation theory. Most researchers choose gene mutation as their research subject, since cancer cell fusion is too difficult to investigate. In this sense, this reviewer strongly suggests publication of this review to attract research interests in cancer cell fusion. On the other hand, this reviewer thinks that current version of the manuscript should be revised to meet the standard of journal publication.
Major comments:
1. On dark matter “phenomenon”.
Dark matter, defined as “a hypothetic form of matter”, is hypothetical and could not be proved by conventional methodology. In this review, dark matter refers to tumor × normal hybrid derived cells, which do exist but are difficult to detect with current cancer research methods.
The authors should decide whether the term “dark matter” is appropriate to be used to describe tumor × normal hybrid derived cells. Should the authors not be confident on the existence of this type of cells, “dark matter hypothesis” should be used. Dark matter per se is hypothetical, and it will never be a phenomenon.
2. On the introduction of dark matter.
The term of dark matter is being used widely in cancer research, referring to various unknowns (e.g., effects of chaos in biomedical alterations, or possible function of non-coding DNA in the genome, etc.). To help readers, the term of “dark matter” in this review should be clearly defined at the beginning of the manuscript. The Abstract introduces this term only with “...the pool of invisible or so-called dark matter TN-hybrid-derived cells inside a tumor…”, but failed to point out who called these cells dark matter.
3. On detection of tumor × normal hybrid derived cells.
According to this review, dark matter tumor × normal hybrid derived cells do exist but are difficult to detect. This reviewer thinks that the review should be more critical in reviewing the published literature. It would be great if the manuscript could change the title of Section 3.1 (into What are the current methods for detecting tumor × normal hybrid derived cells?), and then add a separate section at the end of the review to discuss the following issues:
1) What is the flaw of current detection of tumor × normal hybrid derived cells?
2) Should gene expression or protein marker be used as proof of tumor × normal hybrid derived cells?
3) How to distinguish tumor × normal hybrid derived cells from cancer cell lineage plasticity caused by transdifferentiation, cancer stem cell property, epithelial to mesenchymal transition, mesenchymal to epithelial transition, and tissue cell mimicry?
4) Theoretically, what is the definitive proof of tumor × normal hybrid derived cells? Considering that most cancer patients do not have a transplant history, how the detection of tumor × normal hybrid derived cells could ever be translated into clinical cancer detection and prognosis?
Minor comments:
1. The manuscript should be critically edited for the use of standard English.
2. Enumeration of the sections should be revised.
Recommendation:
Major revision.
Author Response
Reviewer #3
We would like to thank Reviewer #3 for his/her helpful comments and corrections to the manuscript. In the following we will reply to the specific questions that were raised in the manuscript.
Page 2: It also remains unknown what happened to the cancer cells that formed the present tumor in the past—please explain this.
This sentence has been removed from the manuscript.
Page 2: PDX [Has this been defined? I don’t see it].
Yes, this was defined in that part that has been deleted by you. In any case, PDX has been defined in the revised manuscript. Lines 92-93: “However, because the patient-derived xenograft (PDX) cells…“
Page 3: [DOES THIS MEAN THAT THEY WERE IDENTIFIED BY FISH?]
We have critically read the paper of Jacobsen et al. several times.
As stated out in the results section one cell line, called BJ3Z, was obtained from solid PDX tumors. It had a spindle-shaped fibroblastic morphology and was positive for alpha-smooth muscle actin (unknown whether human or murine alpha-SMA), but negative for human cytokeratin 7 suggesting that these cells were probably of mouse stromal origin, and probably had become transformed. Metaphase spreads and karyotypes confirmed the mouse assignment. However, distinct human chromosomes were detected among the mouse chromosomes in about 30% of spreads and karyotype analyses of these cells revealed distinct human banding patterns. Dual-color fluorescence in situ hybridization (FISH) of 42 metaphases showed 26 cells (62%) with all green (mouse) painted chromosomes and 16 (38%) with a combination of red (human) and green (mouse) chromosomes (Jacobsen et al., 2006 Cancer Res 66:8274). In brief, hybrid cells were clearly identified by FISH.
page 4: On the other hand, since chromosome 17’s were not further investigated it remains unclear whether they were of recipient or donor origin. THE PREVIOUS IS INCORRECT. IT WAS A TRISOMY 17 FOUND IN THE PATIENT AND NOT THE Y-CONTAINING DONOR CELLS.
This is correct and this sentence has been removed from the revised manuscript.
page 5: [But what if TN-hybrids will originate that could not be distinguished from non-fused “wildtype” cancer cells? I DON’T UNDERSTAND THIS.]
This sentence has been removed from the revised manuscript.
page 8: Is this what you mean?: Tumor
The legend to Figure 2 was completely revised.
page 10: Cell fusion is not a rare and random event (I thought you said earlier that it is a random event?) but rather a tightly regulated process that has to be both initiated and that has to be terminated.
This sentence was revised (lines 460-461: Cell fusion is a rather a tightly regulated process that has to be both initiated and terminated.)
Round 2
Reviewer 3 Report
The authors addressed all of this reviewer's comments.